# SSDiff: Spatial-spectral Integrated Diffusion Model for Remote Sensing Pansharpening

**Yu Zhong**[†]
University of Electronic Science
and Technology of China
yuuzhong1011@gmail.com

**Xiao Wu**[†]
University of Electronic Science
and Technology of China
wxwsx1997@gmail.com

**Zihan Cao**
University of Electronic Science
and Technology of China
iamzihan666@gmail.com

**Hong-Xia Dou**
Xihua University
hongxiadou1991@126.com

**Liang-Jian Deng**[*]
University of Electronic Science and Technology of China
liangjian.deng@uestc.edu.cn

## Abstract

Pansharpening is a significant image fusion technique that merges the spatial content and spectral characteristics of remote sensing images to generate high-resolution multispectral images. Recently, denoising diffusion probabilistic models have been gradually applied to visual tasks, enhancing controllable image generation through low-rank adaptation (LoRA). In this paper, we introduce a spatial-spectral integrated diffusion model for the remote sensing pansharpening task, called SSDiff, which considers the pansharpening process as the fusion process of spatial and spectral components from the perspective of subspace decomposition. Specifically, SSDiff utilizes spatial and spectral branches to learn spatial details and spectral features separately, then employs a designed alternating projection fusion module (APFM) to accomplish the fusion. Furthermore, we propose a frequency modulation inter-branch module (FMIM) to modulate the frequency distribution between branches. The two components of SSDiff can perform favorably against the APFM when utilizing a LoRA-like branch-wise alternative fine-tuning method. It refines SSDiff to capture component-discriminating features more sufficiently. Finally, extensive experiments on four commonly used datasets, i.e., WorldView-3, WorldView-2, GaoFen-2, and QuickBird, demonstrate the superiority of SSDiff both visually and quantitatively. The code is available at https://github.com/Z-ypnos/SSdiff_main.

## 1 Introduction

Due to physical limitations, satellite sensors cannot directly acquire high-resolution multispectral images (HrMSI). Instead, they can obtain high-resolution panchromatic (PAN) images and low-resolution multispectral images (LrMSI). Pansharpening techniques can merge PAN images with LrMSI, generating HrMSI that possess both high spatial and spectral resolutions. Pansharpening,

---

[†]Equal contribution.
[*]Corresponding author.

38th Conference on Neural Information Processing Systems (NeurIPS 2024).

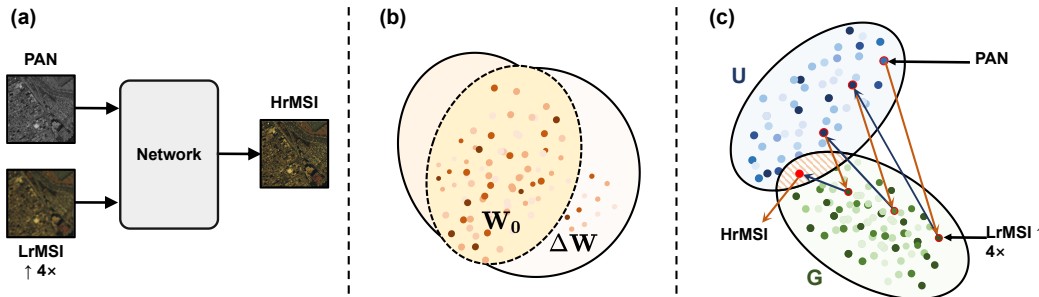

Figure 1: Schematic of (a) DL-based pansharpening approach in a supervised fashion, in which the "network" can be any deep module, e.g., denoising diffusion probabilistic models (DDPM). The comparison of (b) the LoRA based on DDPM and (c) the proposed APFM in our SSDiff. $\mathbf{G}$ and $\mathbf{U}$ represent the spectral and spatial domains, respectively. The LoRA can expand learnable weights $\mathbf{W_0}$ with $\Delta\mathbf{W}$ (but without applications to pansharpening), and the given APFM can obtain pansharpened HrMSI from PAN image and LrMSI through alternating projections.

as a fundamental preprocessing method, has been widely utilized in various applications, including change detection [37] and image segmentation [45].

Pansharpening methods are roughly categorized into four types: component substitution (CS) methods, multi-resolution analysis (MRA) methods, variational optimization (VO) techniques, and deep learning (DL) methods, as shown in Fig. 1 (a). The CS method [17, 22] involves projecting the LrMSI into a specific domain and replacing the spatial components of the LrMSI in that domain with the corresponding components from the PAN image. CS-based methods can generate fusion images with high spatial fidelity but spectral distortion with fast runtime. MRA-based methods [24, 32] extract spatial details from the PAN image through multiscale decomposition and inject them into the LrMSI. While MRA-based methods effectively preserve spectral information, they may sacrifice spatial details. Compared to CS-based and MRA-based methods, VO-based techniques [40, 41] gain more mathematical guarantees but require handling a higher computational burden and more adjustable parameters.

In recent years, DL-based methods [12, 39, 42, 38, 8, 19] have increasingly been applied across various fields, including pansharpening tasks, yielding exciting results. Traditional DL-based methods typically utilize a single-scale model to process information from PAN images and LrMSI. In a single-scale network, PAN images and LrMSI are usually stacked together without distinguishing the information contained within them, serving as inputs to the network. They overlook the disparities in the deep-level information inherent in both, potentially leading to the omission of crucial discriminative features and subsequently influencing lower fusion performance. Then, dual-branch methods [6, 18] based on deep learning can differentiate and hierarchically learn information from PAN and LrMSI. Thanks to this design, it has shown outstanding performance in pansharpening tasks. However, the cumbersome structure of dual-branch networks makes it challenging to perform localized fine-tuning. Denoising diffusion probabilistic model (DDPM) [13] is attaining attention in remote sensing pansharpening [21, 4]. Unfortunately, existing DDPM-based methods have not yet designed models for the discriminative features required in the pansharpening task.

Considering the characteristics of the pansharpening task, we propose a novel SSDiff method based on subspace decomposition, which leverages spatial and spectral branches to discriminatively capture global spatial information and spectral features, respectively. Additionally, we further construct an alternating projection fusion module (APFM) to fuse the captured spatial and spectral components. Besides, a frequency modulation inter-branch module (FMIM) is designed to overcome the problem of uneven distribution of frequency information between two branches in the denoising process. Finally, through the proposed LoRA-like branch-wise alternating fine-tuning (L-BAF), our SSDiff can further reveal spatial and spectral information not discovered in each branch. The main ***contributions*** of this work are as follows:

- Our SSDiff is based on subspace decomposition to divide the network into spatial and spectral branches. In addition, for subspace decomposition, we give an illustration of vector projection and construct an alternating projection fusion module (APFM). APFM transforms the process of fusing HrMSI into the fusion process of spatial and spectral components. Moreover, our SSDiff is tested on four widely used pansharpening datasets and achieves state-of-the-art (SOTA) performance.

- The frequency modulation inter-branch module is used at the junction of spectral and spatial branches to enrich extracted spatial information with more high-frequency information in the denoising process.

- The proposed L-BAF method is used to fine-tune the network based on the proposed APFM, where the spatial and spectral branches are updated alternately. This design allows us to alternately fine-tune the two branches without increasing the parameter count, enabling the learning of more discriminative features.

## 2 Related Works

**DL-based Methods.** As a simple but effective method, the representative single-scale coupling model, namely PNN [20], first proposes a simple and effective three-layer CNN architecture and achieves the best results at that time. Subsequently, other methods such as FusionNet [5], DCFNet [39], and others adopt similar coupled input approaches and successfully design their networks. However, these methods still have significant room for improvement in spectral fidelity and generalization performance due to weak feature representation in their network structure designs. In multi-source image fusion tasks, images acquired from diverse sources exhibit varying characteristics. Coupling two information sources together may suffer from inadequate feature extraction. Then, spatial and spectral branches methods [6, 18, 25] based on deep learning can differentiate and hierarchically learn information from PAN images and LrMSI. These methods can better exploit the potential advantages of multi-scale information.

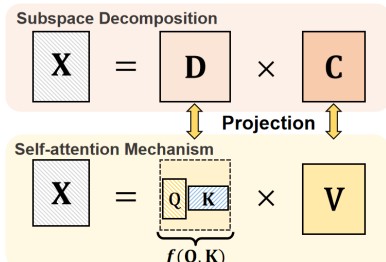

Figure 2: Schematic diagram of the relationship between subspace decomposition and self-attention mechanism. $f(\mathbf{Q}, \mathbf{K})$ is the classic self-similarity equation in self-attention mechanism.

**Diffusion-based Model.** DDPM, as a generative model, has been widely applied in various domains such as text-to-image generation [26], image editing [16] and image classification [43]. In recent years, DDPM has shown its prominence in image processing tasks [9, 28, 44]. Among them, Song et al. [28] propose denoising diffusion implicit models (DDIM), where they design a non-Markov chain sampling process, accelerating the sampling of diffusion models. Then, through some simple modifications, IDDPM [23] achieves competitive log-likelihoods while preserving the high sample quality of DDPM. Currently, DDPM is attracting attention in the field of pansharpening [21, 4]. These DDPM-based methods treat PAN and LrMSI as model fusion conditions, unlike other pansharpening methods where they serve as fusion targets.

**LoRA: Low-rank Adaptation of Large Language Models.** For the fine-tuning of parameters in large pre-trained models, Hu et al. [14] introduce the LoRA to freeze the pre-trained model weights and inject trainable low-rank decomposition matrices into each layer of the Transformer architecture. This significantly reduces the number of trainable parameters for downstream tasks. For $\mathbf{H} = \mathbf{W_0}\mathbf{X}$, the modified forward pass of the LoRA follows the formula:

$$\mathbf{H} = \mathbf{W_0}\mathbf{X} + \Delta\mathbf{W}\mathbf{X} = \mathbf{W_0}\mathbf{X} + \mathbf{B}\mathbf{A}\mathbf{X}, \tag{1}$$

where $\mathbf{W_0} \in \mathbb{R}^{d \times k}$ is a pre-trained weight matrix, $\mathbf{X} \in \mathbb{R}^{k \times n}$, $\mathbf{B} \in \mathbb{R}^{d \times r}$, $\mathbf{A} \in \mathbb{R}^{r \times k}$, and the rank $r \ll \min(d, k)$. Actually, $\mathbf{W_0} + \Delta\mathbf{W} = \mathbf{W_0} + \mathbf{B}\mathbf{A}$ represents a low-rank decomposition.

**Motivation.** PAN images and LrMSI are obtained from different sensors and contain distinct feature information. PAN images exhibit richer spatial details, while LrMSI possesses more abundant spectral information. However, existing DDPM-based methods have not yet designed models specifically for the discriminative features required in the pansharpening task. As a result, these methods suffer from

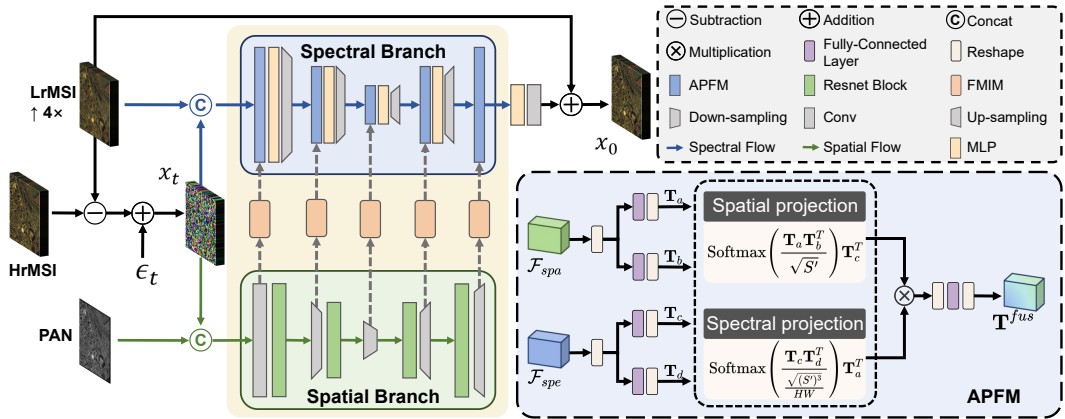

Figure 3: Overall framework of the proposed SSDiff. $\epsilon_t = \sqrt{1 - \bar{\alpha}_t}\epsilon$ is a Gaussian noise, where $t$ is the time step. $\mathcal{F}_{spa}$ is the output of the spatial branch, and $\mathcal{F}_{spe}$ is the output of the spectral branch. The process of APFM follows Theorem 1.

issues such as insufficient feature learning and generalization capabilities, though they may employ low-rank adaptation (LoRA) to improve the performance of DDPM for the pansharpening task.

To alleviate these problems, we propose SSDiff, which transforms the problem of solving HrMSI into a fusion problem of spatial and spectral components. Significantly, we give an illustration of linear algebra to remove the gap between subspace decomposition and the self-attention mechanism. The SSDiff utilizes vector projection to discriminatively capture global spatial information and spectral features in spatial and spectral branches. By introducing subspace decomposition, we can further illustrate and generalize the vector projection to the matrix form. Based on this, we propose an APFM that naturally decouples spatial and spectral information and fuses the captured features. Unlike the LoRA method, the APFM can establish low-rank representations for the spatial-spectral branches more accurately, as shown in Fig. 1. Furthermore, the spectral branch contains abundant low-frequency information. When low- and high-frequency information from the spatial branch is injected into the spectral branch, it may result in an overemphasis on low-frequency information, impacting the denoising performance of the model. Based on this, we propose FMIM to modulate the frequency information between different branches. The overall model is trained by L-BAF to uncover spatial and spectral information not discovered in each branch.

## 3 Methodology

### 3.1 SSDiff Architecture

Inspired by the LoRA approach, we view the pansharpening task as the fusion of spatial and spectral components, where the spatial and spectral elements can be considered as a matrix decomposition of a multi-spectral image. Based on these characteristics, our SSDiff employs a model comprising a spatial branch and a spectral branch, as shown in Fig. 3. Both the spatial branch and the spectral branch comprise two encoder layers and two decoder layers. Down-sampling occurs between the encoder layers to decrease the spatial resolution while increasing the number of channels. Up-sampling operation between the two layers of the decoder increases the spatial resolution while decreasing the number of channels, and a down-sampling convolution layer connects the encoder and the decoder. The spatial branch employs ResNet [11] blocks to convert spatial images into features. These spatial features are transmitted to corresponding layers in the spectral branch via a frequency modulation inter-branch module. Additionally, fusing incoming spatial features and spectral information via an alternating projection fusion module.

Eventually, it is delivered to the next stage via an MLP. In this work, we convert the objective from $\epsilon$ to $x_0$, so the loss function $\mathcal{L}_{simple}$ [4] takes the following form:

$$\mathcal{L}_{simple} = \mathbb{E}\left[\|x_0 - x_\theta(\mathbf{x}_t, \mathbf{c}, t)\|_1\right], \tag{2}$$

where $x_\theta$ denotes the prediction of the model and $\mathbf{c}$ is the conditions for injecting the model. Inspired by FusionNet [5], our SSDiff changes the forward and backward denoising objects during the training process from HrMSI to the difference between HrMSI and up-sampled LrMSI. The detailed training process of SSDiff can be found in Appendix. D.

## 3.2 Alternating Projection Fusion

This section starts with vector projection in linear algebra (See Lemma 1) and subspace decomposition (See Definition 1) to illustrate vector projection as a specific subspace decomposition. Then, the self-attention mechanism [30] is generalized into the proposed alternating projection fusion framework, i.e., APFM. In what follows, we rewrite vector projection as follows.

**Lemma 1** ([29]). *Assuming that the existing two arbitrary vectors $\mathbf{a} \in dom\mathbf{U} \in \mathbb{R}^n$ and $\mathbf{b} \in dom\mathbf{G} \in \mathbb{R}^n$, then $\mathbf{Pb} = \lambda\mathbf{a} = \mathbf{p}$, we have the following formula:*

$$\mathbf{p} = \frac{\mathbf{aa}^T}{\mathbf{a}^T\mathbf{a}}\mathbf{b}. \tag{3}$$

*where $\mathbf{P}$ is projection matrix, $\lambda$ is the scaling factor, and $\mathbf{p}$ is the vector in the same domain as $\mathbf{a}$.*

*Proof.* For any two vectors $\mathbf{a}$ and $\mathbf{b}$, there exists a vector $\mathbf{e} = \mathbf{p} - \mathbf{b}$ such that $\mathbf{e}$ is orthogonal to $\mathbf{a}$. We have the following equation:

$$\mathbf{a}^T\mathbf{e} = \mathbf{a}^T(\mathbf{p} - \mathbf{b}) = \mathbf{a}^T(\lambda\mathbf{a} - \mathbf{b}) = 0, \tag{4}$$

thus, we have

$$\lambda = \frac{\mathbf{a}^T\mathbf{b}}{\mathbf{a}^T\mathbf{a}}. \tag{5}$$

Taking Eq. (5) into $\mathbf{Pb} = \lambda\mathbf{a} = \mathbf{p}$, we have the conclusion:

$$\mathbf{p} = \mathbf{Pb} = \mathbf{a}\lambda = \frac{\mathbf{aa}^T}{\mathbf{a}^T\mathbf{a}}\mathbf{b}. \tag{6}$$

$\square$

**Definition 1** ([7]). *Assume that $\mathbf{D} \in \mathbb{R}^{S \times L}$ is the subspace and $\mathbf{C} \in \mathbb{R}^{L \times HW}$ is the corresponding coefficients. We have:*

$$\mathbf{Z} = \mathbf{DC}. \tag{7}$$

Based on subspace decomposition, we can take spatial and spectral components to accomplish the pansharpening of remote sensing images. According to Lemma 1, we can further determine a specific subspace decomposition in Definition 1. The subspace represents the projection relationship between vectors $\mathbf{a}$ and $\mathbf{b}$. Interestingly, we find that we can generalize Eq. (3) and Eq. (7) to the matrix form of the self-attention mechanism, as shown in Fig. 2. In other words, the self-attention mechanism is represented as the vector projection of Eq. (3), which is the low-rank subspace decomposition.

**Remark 1.** *Back to the pansharpening applications, the input PAN and LrMSI are mapped to a high-dimensional feature space. The features often exhibit significant correlations between frequency bands, while spectral vectors typically reside in a low-dimension subspace. These features can be represented as Eq. (7). In this way, the characteristics in the image domain can be transformed into the subspace.*

Based on the above analysis, we can build an alternating projection framework, which is summarized in the following theorem.

**Theorem 1.** *Assuming that $\mathcal{F}_{spa} \in \mathbb{R}^{H \times W \times S}$ and $\mathcal{F}_{spe} \in \mathbb{R}^{H \times W \times S}$ from the spatial and spectral branches, they can be alternatively projected as follows:*

$$\mathbf{T}^{spa} = Softmax\left(\frac{\mathbf{T}_a\mathbf{T}_b^T}{\sqrt{S'}}\right)\mathbf{T}_c^T, \quad \mathbf{T}^{spe} = Softmax\left(\frac{\mathbf{T}_c\mathbf{T}_d^T}{\frac{\sqrt{(S')^3}}{HW}}\right)\mathbf{T}_a^T, \tag{8}$$

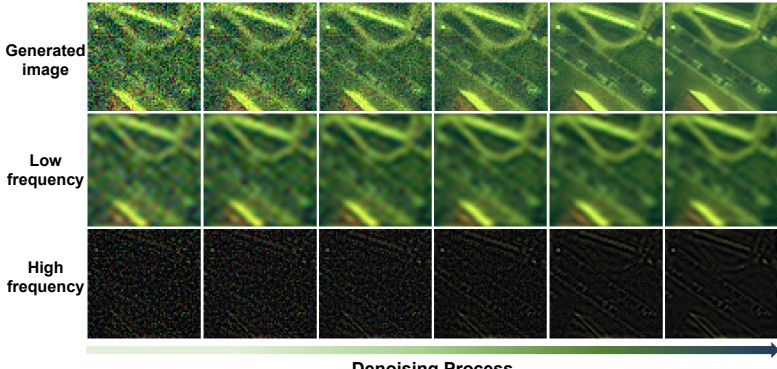

Figure 4: The denoising process. The top row consists of a series of iteratively generated images from the gradual denoising process. The subsequent two rows represent the associated low-frequency and high-frequency spatial domain information obtained through inverse Fourier transform from the denoised image in the first row of each corresponding step.

*where $\mathbf{T}^{spa} \in \mathbb{R}^{HW \times S'}$ and $\mathbf{T}^{spe} \in \mathbb{R}^{S' \times HW}$ denote the features of spatial domain and spectral domain separately. $\mathbf{T}_a \in \mathbb{R}^{HW \times S'}$ and $\mathbf{T}_b \in \mathbb{R}^{HW \times S'}$ are features in the spatial domain generated by $\mathcal{F}_{spa}$, where $S'$ is the channel of self-attention. $\mathbf{T}_c \in \mathbb{R}^{S' \times HW}$ and $\mathbf{T}_d \in \mathbb{R}^{S' \times HW}$ are features in the spectral domain generated by $\mathcal{F}_{spe}$. $\sqrt{S'}$ and $\frac{\sqrt{(S')^3}}{HW}$ are constants related to the matrix size. Softmax($\cdot$) stands for the Softmax function.*

*Proof.* According to Lemma 1, we can generalize the self-attention mechanism [30], where $\mathbf{Q}$ and $\mathbf{K}$ are the features from dom$\mathbf{U}$, and $\mathbf{V}$ is the feature from dom$\mathbf{G}$, respectively. Thus, we have:

$$\text{Softmax}(\frac{\mathbf{Q}\mathbf{K}^T}{\sqrt{d_k}}) \Rightarrow \frac{\mathbf{a}\mathbf{a}^T}{\mathbf{a}^T\mathbf{a}}, \ \mathbf{V} \Rightarrow \mathbf{b}, \tag{9}$$

where "$\Rightarrow$" suggests that the left and right parts of the equation are similar in form, and the left side is a special case of the right side. Through this equation, we can partially explain the principle of cross-attention using the vector projection theorem. Then we transform the projection relationship between the spatial domain (dom$\mathbf{U}$) and spectral domain (dom$\mathbf{G}$), where $\mathbf{T}_a, \mathbf{T}_b \in$ dom$\mathbf{U}$ and $\mathbf{T}_c, \mathbf{T}_d \in$ dom$\mathbf{G}$. $d$ is a self-attention constant. As a result, the alternating projection is complete from the spatial/spectral domain to the spectral/spatial domain, i.e., Eq. (8). $\square$

In addition, we need to get fused outputs from $\mathbf{T}^{spa}$ and $\mathbf{T}^{spe}$. Without loss of generality, we have

$$\mathbf{T}^{fus} = \mathbf{T}^{spa} \odot \mathbf{T}^{spe}, \tag{10}$$

where $\odot$ defines element-wise multiplication. Element-wise multiplication is used to fuse spatial and spectral information to obtain $\mathbf{T}^{fus} \in \mathbb{R}^{HW \times S'}$.

Comparing Eq. (3) with Eq. (8), this subspace is built by vector projection and naturally decouples spatial and spectral information into the self-attention mechanism. This inspires us to apply a fine-tuning method similar to LoRA methods (See details in Sect. 3.4).

### 3.3 Frequency Modulation Inter-branch Module

Through the APFM, we build an effective fusion module from the characteristics of images. Interestingly, there are some differences between the spatial and spectral components. The spectral branch contains abundant low-frequency information. When low- and high-frequency information from the spatial branch is injected into the spectral branch, it may result in an overemphasis on low-frequency information, impacting the denoising performance of the model. We found that modulating the frequency distribution contributes to SSDiff obtaining better fusion results.

As shown in Fig. 4, the low-frequency components undergo a gradual modulation characterized by a slow and subtle rate of change in the denoising process. In contrast, the modulation process of the

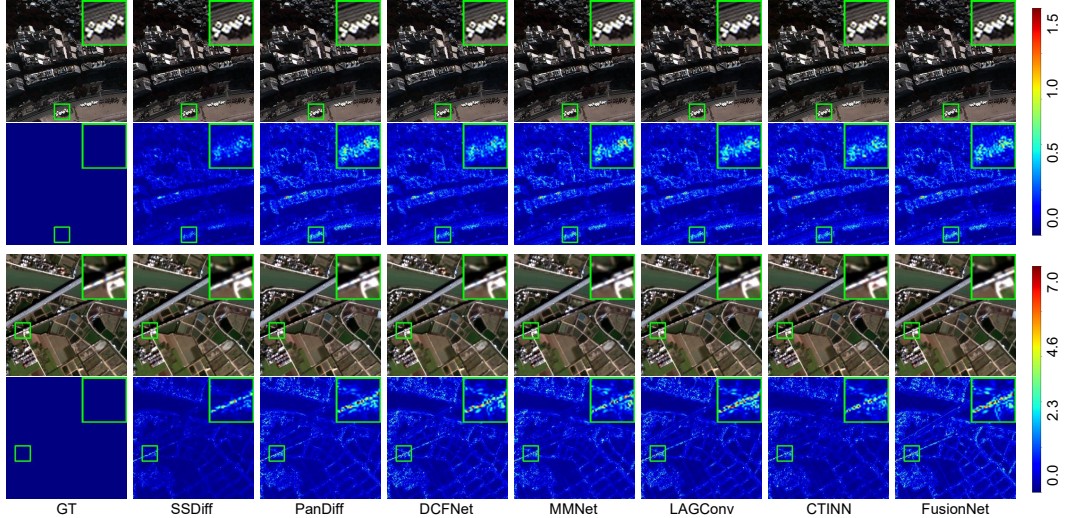

Figure 5: Visual comparisons on a reduced-resolution WorldView-3 and GaoFen-2 case. The first two rows are the results of WV3, and the last two rows are the results of GF2. The first and third rows are the predicted HrMSI for each method, and the second and fourth rows are the error maps of the predicted HRMS versus ground truth (GT) for each one.

high-frequency components exhibits distinct dynamic variation. Considering the above phenomenon, we design a frequency modulation inter-branch module. Specifically, we utilize a Fourier filter to extract the high-frequency information of the feature map $\mathbf{x}_{spa}$ obtained from the spatial branch, following:

$$\mathcal{F}'(\mathbf{x}_{spa}) = \mathbf{FFT}(\mathbf{x}_{spa}) \odot \alpha, \tag{11}$$

$$\mathbf{x}'_{spa} = \mathbf{IFFT}(\mathcal{F}'(\mathbf{x}_{spa})), \tag{12}$$

where $\mathbf{FFT}$ and $\mathbf{IFFT}$ are Fourier transform and inverse Fourier transform. $\odot$ denotes element-wise multiplication, and $\alpha$ is a Fourier mask [27]. Furthermore, we find in experiments that directly injecting high-frequency information into spectral branches will cause the frequency information imbalance. To solve this problem, half of the channels of feature $\mathbf{x}_{spec}$ of the spectral branch are multiplied by a constant. For the three different channel numbers in the model, ranging from low to high, we set this constant to 1.2, 1.4, and 1.6, respectively.

### 3.4 LoRA-like Branch-wise Alternative Fine-tuning

During the model training process, it is crucial to carefully maintain a balance between model underfitting and overfitting. Our SSDiff can hierarchically and discriminatively extract more features. However, achieving a simultaneous balance condition for both spatial and spectral branches during unified training is undoubtedly challenging. Therefore, this paper approaches the LoRA-like alternate fine-tuning of each branch as a feasible solution. As shown in Fig. 1, LoRA methods fine-tune the output of the model by updating the parameters on the fully connected layer weights. Compared with LoRA, the proposed alternating projection method is also a low-rank matrix decomposition. In Fig. 1 (b), the difference is that we can have the backpropagation process and control the gradient of the projection process to achieve alternate fine-tuning in the proposed APFM. In practice, taking the fine-tuning of Eq. (8) ($\mathbf{T}^{spa}$) as an example to update the spectral branch, we can detach the gradient propagation at $\mathbf{T}^{spa}$, preventing parameter updates in the spatial branch. In this case, gradients only propagate through the path shown in Eq. (8) ($\mathbf{T}^{spe}$), which means only the parameters of the spectral branch are updated. Similarly, when fine-tuning the spatial branch, detaching the gradient propagation from the computation graph at $\mathbf{T}^{spe}$ can prevent parameter updates in the spectral branch. A more intuitive fine-tuning process is provided in the supplementary material.

Table 1: Result on the WV3 (the first thirteen rows) and GF2 (the last thirteen rows) reduced-resolution and full-resolution datasets. The best results are highlighted in bold and the second best results are underlined.

| Method | Reduced resolution | | | | Full resolution | | |
|---|---|---|---|---|---|---|---|
| | SAM($\pm$ std) | ERGAS($\pm$ std) | Q2$^n$($\pm$ std) | SCC($\pm$ std) | $D_\lambda$($\pm$ std) | $D_s$($\pm$ std) | HQNR($\pm$ std) |
| BDSD-PC [31] | 5.4675$\pm$1.7185 | 4.6549$\pm$1.4667 | 0.8117$\pm$0.1063 | 0.9049$\pm$0.0419 | 0.0625$\pm$0.0235 | 0.0730$\pm$0.0356 | 0.8698$\pm$0.0531 |
| MTF-GLP-FS [33] | 5.3233$\pm$1.6548 | 4.6452$\pm$1.4441 | 0.8177$\pm$0.1014 | 0.8984$\pm$0.0466 | 0.0206$\pm$0.0082 | 0.0630$\pm$0.0284 | 0.9180$\pm$0.0346 |
| BT-H [1] | 4.8985$\pm$1.3028 | 4.5150$\pm$1.3315 | 0.8182$\pm$0.1019 | 0.9240$\pm$0.0243 | 0.0574$\pm$0.0232 | 0.0810$\pm$0.0374 | 0.8670$\pm$0.0540 |
| PNN [20] | 3.6798$\pm$0.7625 | 2.6819$\pm$0.6475 | 0.8929$\pm$0.0923 | 0.9761$\pm$0.0075 | 0.0213$\pm$0.0080 | 0.0428$\pm$0.0147 | 0.9369$\pm$0.0212 |
| DiCNN [12] | 3.5929$\pm$0.7623 | 2.6733$\pm$0.6627 | 0.9004$\pm$0.0871 | 0.9763$\pm$0.0072 | 0.0362$\pm$0.0111 | 0.0462$\pm$0.0175 | 0.9195$\pm$0.0258 |
| MSDCNN [36] | 3.7773$\pm$0.8032 | 2.7608$\pm$0.6884 | 0.8900$\pm$0.0900 | 0.9741$\pm$0.0076 | 0.0230$\pm$0.0091 | 0.0467$\pm$0.0199 | 0.9316$\pm$0.0271 |
| FusionNet [5] | 3.3252$\pm$0.6978 | 2.4666$\pm$0.6446 | 0.9044$\pm$0.0904 | 0.9807$\pm$0.0069 | 0.0239$\pm$0.0090 | 0.0364$\pm$0.0137 | 0.9406$\pm$0.0197 |
| CTINN [48] | 3.2523$\pm$0.6436 | 2.3936$\pm$0.5194 | 0.9056$\pm$0.0840 | 0.9826$\pm$0.0046 | 0.0550$\pm$0.0288 | 0.0679$\pm$0.0312 | 0.8815$\pm$0.0488 |
| LAGConv [15] | 3.1042$\pm$0.5585 | 2.2999$\pm$0.6128 | 0.9098$\pm$0.0907 | 0.9838$\pm$0.0068 | 0.0368$\pm$0.0148 | 0.0418$\pm$0.0152 | 0.9230$\pm$0.0247 |
| MMNet [49] | 3.0844$\pm$0.6398 | 2.3428$\pm$0.6260 | 0.9155$\pm$0.0855 | 0.9829$\pm$0.0056 | 0.0540$\pm$0.0232 | 0.0336$\pm$0.0115 | 0.9143$\pm$0.0281 |
| DCFNet [39] | 3.0264$\pm$0.7397 | 2.1588$\pm$0.4563 | 0.9051$\pm$0.0881 | 0.9861$\pm$0.0038 | 0.0781$\pm$0.0812 | 0.0508$\pm$0.0342 | 0.8771$\pm$0.1005 |
| PanDiff [21] | 3.2968$\pm$0.6010 | 2.4667$\pm$0.5837 | 0.8980$\pm$0.0880 | 0.9800$\pm$0.0063 | 0.0273$\pm$0.0123 | 0.0542$\pm$0.0264 | 0.9203$\pm$0.0360 |
| SSDiff (ours) | **2.8429$\pm$0.5284** | **2.1059$\pm$0.4560** | **0.9156$\pm$0.0841** | **0.9867$\pm$0.0038** | **0.0132$\pm$0.0049** | **0.0307$\pm$0.0029** | **0.9565$\pm$0.0057** |
| BDSD-PC [31] | 1.7110$\pm$0.3210 | 1.7025$\pm$0.4056 | 0.9932$\pm$0.0308 | 0.9448$\pm$0.0166 | 0.0759$\pm$0.0301 | 0.1548$\pm$0.0280 | 0.7812$\pm$0.0409 |
| MTF-GLP-FS [33] | 1.6757$\pm$0.3457 | 1.6023$\pm$0.3545 | 0.8914$\pm$0.0256 | 0.9390$\pm$0.0197 | 0.0336$\pm$0.0129 | 0.1404$\pm$0.0277 | 0.8309$\pm$0.0334 |
| BT-H [1] | 1.6810$\pm$0.3168 | 1.5524$\pm$0.3642 | 0.9089$\pm$0.0292 | 0.9508$\pm$0.0150 | 0.0602$\pm$0.0252 | 0.1313$\pm$0.0193 | 0.8165$\pm$0.0305 |
| PNN [20] | 1.0477$\pm$0.2264 | 1.0572$\pm$0.2355 | 0.9604$\pm$0.0100 | 0.9772$\pm$0.0054 | 0.0367$\pm$0.0291 | 0.0943$\pm$0.0224 | 0.8726$\pm$0.0373 |
| DiCNN [12] | 1.0525$\pm$0.2310 | 1.0812$\pm$0.2510 | 0.9594$\pm$0.0101 | 0.9771$\pm$0.0058 | 0.0413$\pm$0.0128 | 0.0992$\pm$0.0131 | 0.8636$\pm$0.0165 |
| MSDCNN [36] | 1.0472$\pm$0.2210 | 1.0413$\pm$0.2309 | 0.9612$\pm$0.0108 | 0.9782$\pm$0.0050 | 0.0269$\pm$0.0131 | 0.0730$\pm$0.0093 | 0.9020$\pm$0.0128 |
| FusionNet [5] | 0.9735$\pm$0.2117 | 0.9878$\pm$0.2222 | 0.9641$\pm$0.0093 | 0.9806$\pm$0.0049 | 0.0400$\pm$0.0126 | 0.1013$\pm$0.0134 | 0.8628$\pm$0.0184 |
| CTINN [48] | 0.8251$\pm$0.1386 | 0.6995$\pm$0.1068 | 0.9772$\pm$0.0117 | 0.9803$\pm$0.0015 | 0.0586$\pm$0.0260 | 0.1096$\pm$0.0149 | 0.8381$\pm$0.0237 |
| LAGConv [15] | 0.7859$\pm$0.1478 | 0.6869$\pm$0.1125 | 0.9804$\pm$0.0085 | 0.9906$\pm$0.0019 | 0.0324$\pm$0.0130 | 0.0792$\pm$0.0136 | 0.8910$\pm$0.0204 |
| MMNet [49] | 0.9929$\pm$0.1411 | 0.8117$\pm$0.1185 | 0.9690$\pm$0.0204 | 0.9859$\pm$0.0024 | 0.0428$\pm$0.0300 | 0.1033$\pm$0.0129 | 0.8583$\pm$0.0269 |
| DCFNet [39] | 0.8896$\pm$0.1577 | 0.8061$\pm$0.1369 | 0.9727$\pm$0.0100 | 0.9853$\pm$0.0024 | 0.0234$\pm$0.0116 | 0.0659$\pm$0.0096 | 0.9122$\pm$0.0119 |
| PanDiff [21] | 0.8881$\pm$0.1197 | 0.7461$\pm$0.1032 | 0.9792$\pm$0.0097 | 0.9887$\pm$0.0020 | 0.0265$\pm$0.0195 | 0.0729$\pm$0.0103 | 0.9025$\pm$0.0209 |
| SSDiff (ours) | **0.6694$\pm$0.1244** | **0.6038$\pm$0.1080** | **0.9836$\pm$0.0074** | **0.9915$\pm$0.0017** | **0.0164$\pm$0.0093** | **0.0267$\pm$0.0071** | **0.9573$\pm$0.0100** |
| Ideal value | **0** | **0** | **1** | **1** | **0** | **0** | **1** |

## 4 Experiments

### 4.1 Experimental Results

**Results on WorldView-3.** On the WorldView-3 dataset, we evaluate the performance of SSDiff using 20 test images. The results for both reduced-resolution and full-resolution are presented in Table 1. The running time of a single picture during the sampling process is 7.417 seconds under 10 timesteps. We compared our method with three traditional methods and some SOTA DL-based methods. To illustrate the performance of each method more clearly, we presented the fusion result images and error maps of some of these methods in Fig. 5, and zoomed in on a specific location. On average, our method achieves SOTA performance on the reduced dataset, with our SSDiff reaching 2.84 (SAM) and 2.10 (ERGAS) metrics, outperforming all DL-based methods. The error map indicates that the images sampled by SSDiff are closer to the ground truth (GT). SSDiff achieves SOTA performance in obtaining full-resolution images on the WV3 dataset. The HQNR score close to 1 indicates better fusion quality of the full-resolution images. The obtained results demonstrate that our SSDiff can fuse HrMSI, reducing spatial and spectral distortions, thereby proving its excellent generalization ability at full resolution.

**Results on GaoFen-2.** On the GaoFen-2 reduced dataset, we tested our SSDiff on 20 test images, as shown in Table 1. Our SSDiff achieves SOTA performance. From the error maps in Fig. 5, we can observe that there are still significant differences between traditional fusion methods and DL-based fusion methods. These experiments demonstrate that, compared to other DL-based methods used for comparison in the experiments, the proposed SSDiff exhibits superior spatial performance and effectively preserves spectral information.

### 4.2 Ablation Study

**Effectiveness of Decoupling Branches.** To investigate the effectiveness of the spatial and spectral branch design, we perform ablation experiments by training the diffusion model under the following

conditions: V1) Coupling inputs, and training only with the spatial branch. V2) Coupling inputs and training only with the spectral branch. V3) Spatial and spectral branch structure and decoupling inputs, only the outputs of the two branches are concatenated without any inter-branch information interaction. V4) Spatial and spectral branch structure and decoupling inputs, replace each APFM with an additional operation, i.e., the way of LoRA. V5) Spatial and spectral branch structure and decoupling inputs, replace each APFM with a multiplication operation. V6) Our method. The results on the WV3 reduced dataset are reported in Table 2. Training solely with a single branch significantly reduces the quality of the HrMSI. It can be seen that SSDiff is more suitable for pansharpening tasks than the LoRA way, multiplication, and concatenating way.

Table 2: Ablation study on 20 reduced-resolution samples from WV3 dataset without fine-tuning.

| Method | SAM($\pm$ std) | ERGAS($\pm$ std) | Q2$^n$($\pm$ std) | SCC($\pm$ std) | Params |
|--------|---------------|------------------|-------------------|----------------|--------|
| V1 | 3.3612$\pm$0.6497 | 2.5633$\pm$0.6249 | 0.8960$\pm$0.1031 | 0.9816$\pm$0.0070 | 1100K |
| V2 | 3.0598$\pm$0.5560 | 2.2638$\pm$0.5663 | 0.9097$\pm$0.0947 | 0.9847$\pm$0.0062 | 654K |
| V3 | 3.4040$\pm$0.6088 | 2.5564$\pm$0.6737 | 0.9023$\pm$0.0954 | 0.9796$\pm$0.0078 | 1420K |
| V4 | 3.3245$\pm$0.5342 | 2.4371$\pm$0.5841 | 0.9089$\pm$0.0878 | 0.9830$\pm$0.0069 | 1420K |
| V5 | 3.1958$\pm$0.5727 | 2.3962$\pm$0.5688 | 0.9035$\pm$0.0962 | 0.9834$\pm$0.0067 | 1420K |
| V6 | **2.8646$\pm$0.5241** | **2.1217$\pm$0.4671** | **0.9125$\pm$0.0874** | **0.9863$\pm$0.0040** | 1420K |

**Frequency Modulation Inter-branch Module.** To validate the effectiveness of the FMIM, we remove FMIM from our model and train the diffusion model to converge the WV3 dataset. The results are shown in Table 3. Without using FMIM, the model's performance on the SAM/ERGAS/Q8 indicators decreased by approximately 3.1%/2.8%/1%, respectively. This demonstrates that utilizing FMIM for frequency transfer can effectively improve model performance.

Table 3: Ablation study on 20 reduced-resolution samples from WV3 dataset without fine-tuning.

| FMIM | SAM($\pm$ std) | ERGAS($\pm$ std) | Q2$^n$($\pm$ std) | SCC($\pm$ std) |
|------|---------------|------------------|-------------------|----------------|
| ✗ | 2.9798$\pm$0.6060 | 2.1978$\pm$0.5399 | 0.9153$\pm$0.0868 | 0.9855$\pm$0.0053 |
| ✓ | **2.8646$\pm$0.5241** | **2.1217$\pm$0.4671** | **0.9125$\pm$0.0874** | **0.9863$\pm$0.0040** |

## 4.3 Discussion

**Generalization.** To test the generalization ability of DL-based methods, we evaluated models trained on the WV3 dataset using 20 reduced resolutions from the WorldView-2 dataset. The quantitative evaluation results, as reported in Table 4, demonstrate that the SSDiff method achieves the best results across all four evaluation metrics. This indicates that our approach possesses a powerful generalization ability.

Table 4: Generalization of DL-based methods on WV2 dataset.

| Method | SAM($\pm$ std) | ERGAS($\pm$ std) | Q2$^n$($\pm$ std) | SCC($\pm$ std) |
|--------|---------------|------------------|-------------------|----------------|
| PNN | 7.1158$\pm$1.6812 | 5.6152$\pm$0.9431 | 0.7619$\pm$0.0928 | 0.8782$\pm$0.0175 |
| DiCNN | 6.9216$\pm$0.7898 | 6.2507$\pm$0.5745 | 0.7205$\pm$0.0746 | 0.8552$\pm$0.0289 |
| MSDCNN | 6.0064$\pm$0.6377 | 4.7438$\pm$0.4939 | 0.8241$\pm$0.0799 | 0.8972$\pm$0.0109 |
| FusionNet | 6.4257$\pm$0.8602 | 5.1363$\pm$0.5151 | 0.7961$\pm$0.0737 | 0.8746$\pm$0.0134 |
| CTINN | 6.4103$\pm$0.5953 | 4.6435$\pm$0.3792 | 0.8172$\pm$0.0873 | 0.9147$\pm$0.0102 |
| LAGConv | 6.9545$\pm$0.4739 | 5.3262$\pm$0.3185 | 0.8054$\pm$0.0837 | 0.9125$\pm$0.0101 |
| MMNet | 6.6109$\pm$0.3209 | 5.2213$\pm$0.2133 | 0.8143$\pm$0.0790 | 0.9136$\pm$0.0201 |
| DCFNet | 5.6194$\pm$0.6039 | 4.4887$\pm$0.3764 | 0.8292$\pm$0.0815 | 0.9154$\pm$0.0083 |
| SSDiff (ours) | **5.0647$\pm$0.5634** | **3.9885$\pm$0.4297** | **0.8577$\pm$0.0782** | **0.9335$\pm$0.0055** |

**Training SSDiff.** To address the issue of insufficient local parameter training in the dual branches model, we design the L-BAF method to fine-tune the model. Taking the experiments on the reduced WV3 dataset, we first train the model without fine-tuning until convergence and perform branch-wise fine-tuning, which includes: 1) Full parameter fine-tuning without L-BAF. 2) Only fine-tuning the spatial branch parameters with the spectral branch parameters fixed. 3) Only fine-tuning the spectral

branch parameters with the spatial parameters fixed. 4) Alternating fine-tuning spectral branch and spatial branch. The quantitative results are shown in Table 5, and all three branch-wise fine-tuning methods lead to improved testing performance. The results of full fine-tuning show a decrease in some metrics, indicating that the method suffers from overfitting. The alternating fine-tuning approach showed the most significant performance improvement. This demonstrates the effectiveness of the L-BAF.

Table 5: Fine-tune SSDiff. $\mathcal{S}$ and $\mathcal{F}$ denote fine-tuning of spatial and spectral branches, respectively.

| $\mathcal{S}$ | $\mathcal{F}$ | SAM($\pm$ std) | ERGAS($\pm$ std) | $Q2^n$($\pm$ std) | SCC($\pm$ std) |
|---|---|---|---|---|---|
| - | - | 2.8646$\pm$0.5241 | 2.1217$\pm$0.4671 | 0.9125$\pm$0.0874 | 0.9863$\pm$0.0040 |
| ✗ | ✗ | 2.8681$\pm$0.5837 | 2.1302$\pm$0.5235 | **0.9206$\pm$0.0850** | **0.9868$\pm$0.0048** |
| ✓ | ✗ | 2.8545$\pm$0.5244 | 2.1138$\pm$0.4658 | 0.9143$\pm$0.0857 | 0.9864$\pm$0.0040 |
| ✗ | ✓ | 2.8460$\pm$0.5232 | 2.1132$\pm$0.4671 | 0.9152$\pm$0.0849 | 0.9864$\pm$0.0041 |
| ✓ | ✓ | **2.8429$\pm$0.5284** | **2.1059$\pm$0.4560** | 0.9156$\pm$0.0841 | 0.9867$\pm$0.0038 |

## 5   Conclusion

In this paper, we propose a spatial-spectral integrated diffusion model for remote sensing pansharpening, named SSDiff. We design a spatial-spectral integrated model architecture, which utilizes spatial and spectral branches to learn spatial details and spectral features separately. By introducing vector projection, the spatial and spectral components in the subspace decomposition are further specified in the proposed APFM. Then, the self-attention mechanism is naturally generalized to the APFM. Furthermore, we propose an FMIM to modulate the frequency distribution between branches. Finally, the two branches of SSDiff can capture discriminating features. It is interesting that, when utilizing the proposed L-BAF method in the APFM, the two branches can be updated alternately, and then SSDiff produces more satisfactory results. We compare our SSDiff with several SOTA pansharpening methods on the WorldView-3, QuickBird, GaoFen-2, and WorldView-2 datasets. The results demonstrate the superiority of SSDiff both visually and quantitatively.

## 6   Acknowledgement

This work is supported by the National Natural Science Foundation of China under Grants 12271083, 12171072 and Natural Science Foundation of Sichuan Province under Grants 2023NSFSC1341.

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

# A  Appendix / supplemental material

In the supplementary material, we first introduce the background knowledge, i.e., denoising diffusion probabilistic models, and then illustrate the process of model training and LoRA-like branch-wise alternative fine-tuning, the limitations, and the broader impact of our method. After that, we provide more details for experiments on the pansharpening task, i.e., Implementation Details, Datasets, Benchmarks, and Quality Metrics. Finally, we display more qualitative evaluation results on the QuickBird and GaoFen-2 datasets.

# B  Background

## B.1  Denoising Diffusion Probabilistic Models

Denoising diffusion probabilistic models [13] are latent variable models, generating realistic target images progressively from a normal distribution by iterative denoising. The diffusion model contains two steps: forward and reverse processes.

The forward process aims to make the prior data distribution $\mathbf{x_0}$ noisy by a $T$ step Markov chain that gradually transforms it into an approximate standard normal distribution $\mathbf{x}_T \sim \mathcal{N}(0, \mathbf{I_d})$ and $\mathbf{d}$ denotes the dimension. One forward step is defined as follows:

$$q(\mathbf{x}_t|\mathbf{x}_{t-1}) = \mathcal{N}(\mathbf{x}_t; \sqrt{1 - \beta_t}\mathbf{x}_{t-1}, \beta_t\mathbf{I}), \tag{13}$$

where $\mathcal{N}(\cdot)$ is a Gaussian distribution with the mean of $\sqrt{1 - \beta_t}\mathbf{x}_{t-1}$ and variance of $\beta_t\mathbf{I}$, $\beta_t$ is a pre-defined variance schedule in time step $t \in [0, T]$. Through the reparameterization trick, we can derive $\mathbf{x}_t$ directly from $\mathbf{x}_0$, The following equation gives this derivation:

$$q(\mathbf{x}_t|\mathbf{x}_0) = \sqrt{\bar{\alpha}_t}\mathbf{x}_0 + \sqrt{1 - \bar{\alpha}_t}\epsilon, \tag{14}$$

where $\epsilon \sim \mathcal{N}(0, \mathbf{I})$ and $\alpha_t = 1 - \beta_t$, $\bar{\alpha}_t = \prod_{i=0}^{t}\alpha_i$.

The reverse process aims to learn to remove the degradation brought from the forward process and sample the $\mathbf{x}_0$ from $\mathbf{x}_t$. To accomplish this objective, we need to learn the distribution of $p_\theta(\mathbf{x}_{t-1}|\mathbf{x}_t)$ using a neural network and perform iterative sampling as follows:

$$p_\theta(\mathbf{x}_{t-1}|\mathbf{x}_t) = \mathcal{N}(\mathbf{x}_{t-1}; \mu_\theta(\mathbf{x}_t, t), \Sigma_\theta(\mathbf{x}_t, t)), \tag{15}$$

where $\mu_\theta$ and $\Sigma_\theta$ are the mean and variance of $p_\theta(\mathbf{x}_{t-1}|\mathbf{x}_t)$, respectively, and $\theta$ is the parameters of model.

According to Eq. (15), the mean and variance can be computed, following:

$$\mu_\theta = \frac{1}{\sqrt{\alpha_t}}(\mathbf{x}_t - \frac{\beta_t}{\sqrt{1 - \bar{\alpha}_t}}\epsilon_\theta(\mathbf{x}_t, t)), \tag{16}$$

$$\Sigma_\theta(\mathbf{x}_t, t) = \frac{1 - \bar{\alpha}_{t-1}}{1 - \bar{\alpha}_t}\beta_t. \tag{17}$$

For sampling from a standard Gaussian noise $\mathbf{x}_T$ to get $\mathbf{x}_{T-1}$, after performing $T$-step iterations of sampling as described above, we get the output $\mathbf{x}_0$ from $\mathbf{x}_T$.

# C  LoRA-like Branch-wise Alternative Fine-tuning

In Section 3.4 of the main text. The L-BAF method alternately fine-tunes the spatial and spectral branches based on the proposed APFM. As shown in Fig. 6. When we fine-tune the spectral branch, we freeze the $\mathbf{T}_a$ and $\mathbf{T}_b$ parameters in the spatial branch and the $\mathbf{T}_a$ parameter in the spectral branch. Block gradient propagation to prevent parameter updates for the entire spatial branch. Similarly, when fine-tuning the spatial branch, we freeze the $\mathbf{T}_c$ and $\mathbf{T}_d$ parameters in the spectral branch, as well as the $\mathbf{T}_c$ parameter in the spatial branch. Block gradient propagation to prevent parameter updates for entire spectral branches.

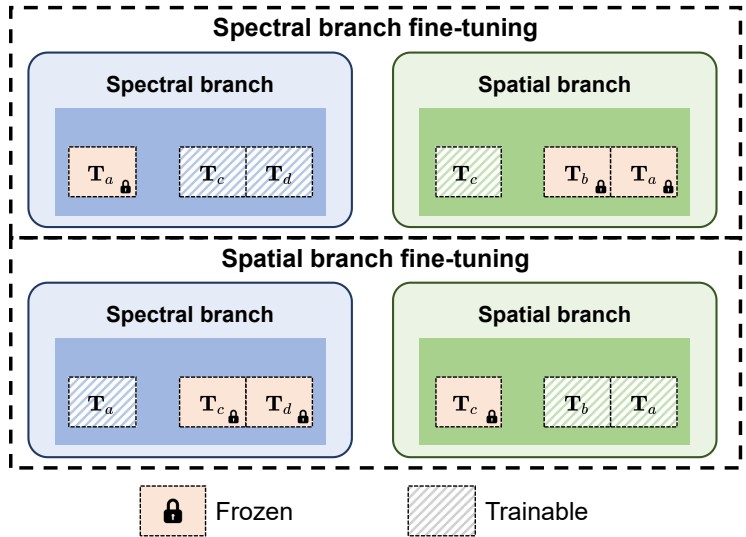

Figure 6: The sketch of the proposed LoRA-like branch-wise alternative fine-tuning process.

---

**Algorithm 1:** Training stage of the proposed method.

---

**Data:** GT image $\mathbf{x}_0$, diffusion model $\mathbf{x}_\theta$ with its parameters $\theta$, spectral and spatial branch
      parameter $\theta_{spe}$, $\theta_{spa}$, respectively, condition $cond$, timestep $t$, and denoised objective $\hat{\mathbf{x}}_0$.

**Result:** Optimized diffusion model $\mathbf{x}_\theta^*$.

1   $\mathbf{cond} \leftarrow \text{PAN}, \text{LrMSI}, \mathbf{x_t}$;

2   **while** *until convergence* **do**

3      $t \leftarrow \text{Uniform}(0, T)$; $\epsilon \sim \mathcal{N}(0, \boldsymbol{I})$;

4      $\mathbf{x}_t \leftarrow \sqrt{\bar{\alpha}_t}(\mathbf{x}_0 - \text{LrMSI}) + \sqrt{1 - \bar{\alpha}_t}\epsilon$; $\hat{\mathbf{x}}_0 \leftarrow \mathbf{x}_\theta(\mathbf{x}_t, \mathbf{cond}) + \text{LrMSI}$;

5      **if** *iteration > 150k* **then**

6          fine-tune $\theta_{spe}$ or $\theta_{spa}$; // L-BAF

7      **end**

8      $\theta \leftarrow \nabla_\theta \mathcal{L}_{simple}(\hat{\mathbf{x}}_0, \mathbf{x}_0)$.

9   **end**

---

## D    Model Training

The detailed training process of SSDiff can be found in Algorithm 1.

## E    Limitation

First, we evaluate the effectiveness of the proposed SSDiff over the pansharpening task and we
will extend our method to other multispectral fusion tasks, such as multispectral and hyperspectral
image fusion. Second, the proposed FMIM can adjust the frequency information between the two
branches, but it also introduces additional hyperparameters, increasing the difficulty of fine-tuning
during training. In terms of cost, the average inference time for a single image is 7.416 seconds under
10 timesteps. The time cost for our approach is higher than other DL-based models, primarily due to
the limitation imposed by the large number of sampling steps required in the diffusion model.

## F    Broader Impact

Pansharpening, as a fundamental preprocessing method which is a key pre-processing technology
overcoming the constraints of hardware before using high-resolution multispectral images, has been
widely utilized in various applications, including change detection, environmental monitoring, and
segmentation. At the same time, pansharpening, being a low-level task, the distortions that occur

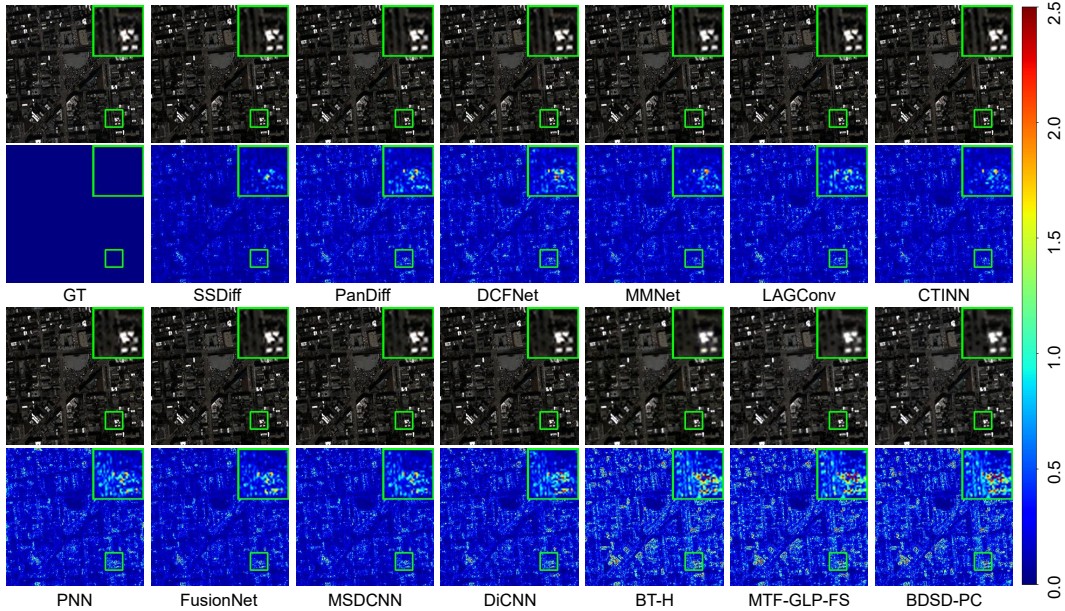

Figure 7: Visual comparisons on a reduced-resolution QuickBird case. The first and third rows are the predicted HrMSI for each method, and the second and fourth rows are the error maps of the predicted HRMS versus ground truth (GT) for each one.

Table 6: Quantitative results on the QuickBird reduced-resolution and full-resolution datasets. Some conventional methods (the first three rows) and DL-based methods are compared. The best results are highlighted in bold and the second best results are underlined.

| Method | Reduced resolution | | | | Full resolution | | |
| | SAM($\pm$ std) | ERGAS($\pm$ std) | Q4($\pm$ std) | SCC($\pm$ std) | $D_\lambda$($\pm$ std) | $D_s$($\pm$ std) | HQNR($\pm$ std) |
|---|---|---|---|---|---|---|---|
| BDSD-PC [31] | 8.2620$\pm$2.0497 | 7.5420$\pm$0.8138 | 0.8323$\pm$0.1013 | 0.9030$\pm$0.0181 | 0.1975$\pm$0.0334 | 0.1636$\pm$0.0483 | 0.6722$\pm$0.0577 |
| MTF-GLP-FS [33] | 8.1131$\pm$1.9553 | 7.5102$\pm$0.7926 | 0.8296$\pm$0.0905 | 0.8998$\pm$0.0196 | 0.0489$\pm$0.0149 | 0.1383$\pm$0.0238 | 0.8199$\pm$0.0340 |
| BT-H [1] | 7.1943$\pm$1.5523 | 7.4008$\pm$0.8378 | 0.8326$\pm$0.0880 | 0.9156$\pm$0.0152 | 0.2300$\pm$0.0718 | 0.1648$\pm$0.0167 | 0.6434$\pm$0.0645 |
| PNN [20] | 5.2054$\pm$0.9625 | 4.4722$\pm$0.3734 | 0.9180$\pm$0.0938 | 0.9711$\pm$0.0123 | 0.0569$\pm$0.0112 | 0.0624$\pm$0.0239 | 0.8844$\pm$0.0304 |
| DiCNN [12] | 5.3795$\pm$1.0266 | 5.1354$\pm$0.4876 | 0.9042$\pm$0.0942 | 0.9621$\pm$0.0133 | 0.0920$\pm$0.0143 | 0.1067$\pm$0.0210 | 0.8114$\pm$0.0310 |
| MSDCNN [36] | 5.1471$\pm$0.9342 | 4.3828$\pm$0.3400 | 0.9188$\pm$0.0966 | 0.9689$\pm$0.0121 | 0.0602$\pm$0.0150 | 0.0667$\pm$0.0289 | 0.8774$\pm$0.0388 |
| FusionNet [5] | 4.9226$\pm$0.9077 | 4.1594$\pm$0.3212 | 0.9252$\pm$0.0902 | 0.9755$\pm$0.0104 | 0.0586$\pm$0.0189 | 0.0522$\pm$0.0088 | 0.8922$\pm$0.0219 |
| CTINN [48] | 4.6583$\pm$0.7755 | 3.6969$\pm$0.2888 | 0.9320$\pm$0.0072 | 0.9829$\pm$0.0072 | 0.1738$\pm$0.0332 | 0.0731$\pm$0.0237 | 0.7663$\pm$0.0432 |
| LAGConv [15] | 4.5473$\pm$0.8296 | 3.8259$\pm$0.4196 | 0.9335$\pm$0.0878 | 0.9807$\pm$0.0091 | 0.0844$\pm$0.0238 | 0.0676$\pm$0.0136 | 0.8536$\pm$0.0178 |
| MMNet [49] | 4.5568$\pm$0.7285 | 3.6669$\pm$0.3036 | 0.9337$\pm$0.0941 | **0.9829$\pm$0.0070** | 0.0890$\pm$0.0512 | 0.0972$\pm$0.0382 | 0.8225$\pm$0.0319 |
| DCFNet [39] | 4.5383$\pm$0.7397 | 3.8315$\pm$0.2915 | 0.9325$\pm$0.0903 | 0.9741$\pm$0.0101 | 0.0454$\pm$0.0147 | 0.1239$\pm$0.0269 | 0.8360$\pm$0.0158 |
| PanDiff [21] | 4.5754$\pm$0.7359 | 3.7422$\pm$0.3099 | 0.9345$\pm$0.0902 | 0.9818$\pm$0.0902 | 0.0587$\pm$0.0223 | 0.0642$\pm$0.0252 | 0.8813$\pm$0.0417 |
| SSDiff (ours) | **4.4640$\pm$0.7473** | **3.6320$\pm$0.2749** | **0.9346$\pm$0.0943** | 0.9829$\pm$0.0080 | **0.0314$\pm$0.0108** | **0.0360$\pm$0.0133** | **0.9338$\pm$0.0208** |
| Ideal value | **0** | **0** | **1** | **1** | **0** | **0** | **1** |

during the process of generating high-resolution multispectral images can significantly impact the success rate of subsequent high-level tasks.

# G   Additional experiment

## G.1   Setups

**Implementation Details.** Our SSDiff is implemented in PyTorch 1.7.0 and Python 3.8.5 using AdamW optimizer with an initial learning rate of 0.001 to minimize $\mathcal{L}_{simple}$ on a Linux operating system with an Intel 12th Gen i7-12700K processor and two NVIDIA GeForce RTX3090 GPUs. For the diffusion denoising model, the initial number of model channels is 32, the diffusion time step used for training in the pansharpening is set to 1000, while the diffusion time step for sampling is set to 10. The exponential moving average (EMA) ratio is set to 0.9999. The total training iterations for the WV3, GF2, and QB datasets are set to 150k, 100k, and 200k iterations, respectively. During the

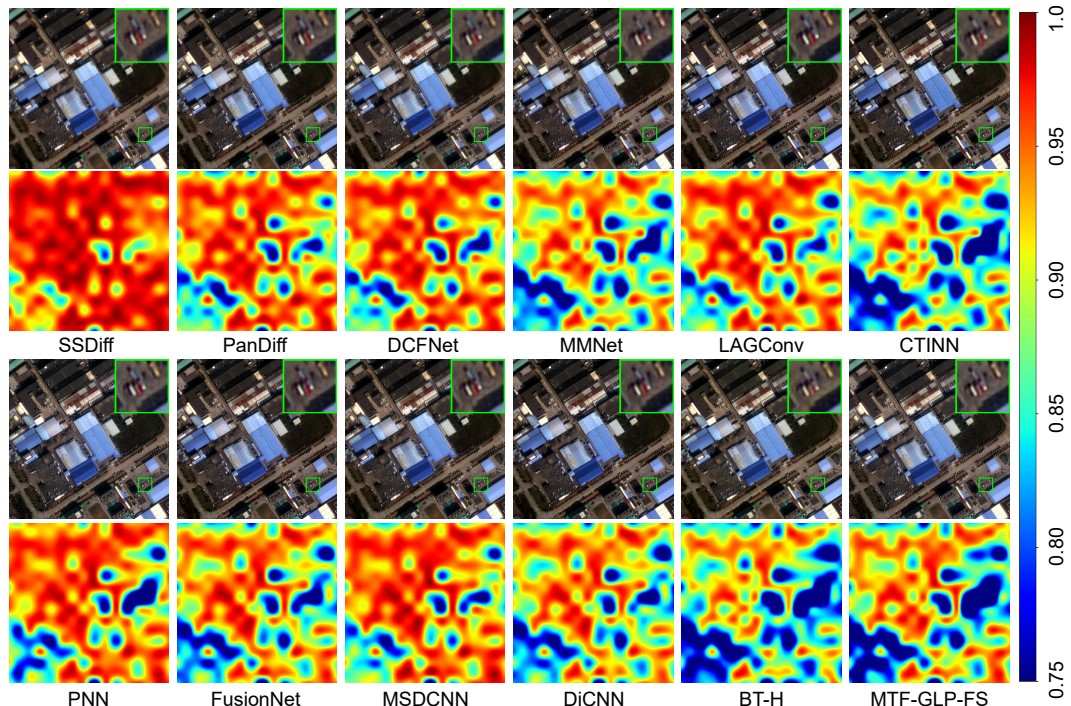

Figure 8: Fused GF2 full-resolution data and their corresponding HQNR map. The high value in the HQNR map means better full-resolution fusion performance.

model fine-tuning, the learning rate is set to 0.0001, and the total fine-tune training iterations are set to 30k.

**Datasets.** To demonstrate the effectiveness of our SSDiff, we conducted experiments on widely used pansharpening datasets. The Pancollection[1] dataset for pansharpening consists of data from four satellites: WorldView-3 (8 bands), WorldView-2 (8 bands), QuickBird (4 bands), and GaoFen-2 (4 bands).

To better evaluate performance, we simulated reduced-resolution and full-resolution datasets. For reduced datasets, Pancollection follows Wald's protocol [35] to obtain simulated images with ground truth images. There are three steps involved: 1) Use a modulation transfer-based (MTF) filter to downsample the original PAN and MS images by a factor of 4. Downsampled PAN and MS are used as training PAN and MS images; 2) Treat the original MS image as a ground truth image, i.e. HRMSI; 3) Upsampling the training MS image using interpolation with polynomial kernels [2] and processing it into a LrMSI. When processing the full datasets, the original MS image is considered MS, the upsampled MS image is considered LrMSI, and the original PAN is considered PAN.

**Benchmark.** To evaluate the performance of our SSDiff, we compared it with various state-of-the-art methods of Pansharpening (on WV3, QB, and GF2 datasets). Specifically, we choose three traditional methods: BDSD-PC [31], MTF-GLP-FS [33], BT-H [1]; as well as nine machine learning-based methods: PNN [20], DiCNN [12], MSDCNN [36], FusionNet [5], CTINN [48], LAGConv [15], MMNet [49], DCFNet [39], and Pandiff [21]. To ensure fairness, we train DL-based methods using the same Nvidia GPU-3090 and PyTorch environment.

**Quality Metric.** For the reduced data in Pansharpening tasks, we utilize four metrics to evaluate the results on reduced resolution datasets, including the spectral angle mapper (SAM) [46], the erreur relative globale adimensionnelle de synthèse (ERGAS) [34], the universal image quality index ($Q2^n$) [10], and the spatial correlation coefficient (SCC) [47]. As for full-resolution datasets, we apply $D_\lambda$, $D_s$, and hybrid quality with no reference (HQNR) indexes [3] for evaluation.

---

[1] https://liangjiandeng.github.io/PanCollection.html.

## G.2 Results on QuickBird:

We conduct experiments on the QuickBird reduced dataset and evaluate the performance of SSDiff. Similarly, the reference and non-reference metrics were obtained from 20 randomly selected test images from the QB dataset. The performance comparison is reported in Table 6. Our SSDiff achieves SOTA performance. From the error maps in Fig. 7, we can observe that there are still significant differences between traditional fusion methods and DL-based fusion methods.

## G.3 Results on GaoFen-2:

On the GaoFen-2 full-resolution dataset, we tested our SSDiff on 20 test images, Fig. 8 shows the results and HQNR maps, where an HQNR score close to 1 indicates better fusion quality of full-resolution images. The obtained results indicate that our SSDiff has a good generalization of the full-resolution dataset.

Table 7: Efficiency results on the WV3 reduced-resolution datasets.

| Method | SSDiff | PanDiff | DCFNet | MMNet | LAGConv |
|---|---|---|---|---|---|
| Runtime (s) | 7.417 | 261.410 | 0.548 | 0.348 | 1.381 |

## G.4 Efficiency analysis

The diffusion-based method generally has more running time than CNN-based methods due to the multiple timesteps of the diffusion mechanism. The comparison of inference running time shown in Table 7 ensures this point. However, *for a fair comparison* with another diffusion-based method for pansharpening, i.e., PanDiff, our method still gets a significant advantage.

