# OpenReview forum: "SSDiff: Spatial-spectral Integrated Diffusion Model for Remote Sensing Pansharpening"
_NeurIPS.cc/2024/Conference — NeurIPS 2024 poster_

### Official Review · Reviewer_qNHH · 2024-07-05

**Soundness:** 3
**Presentation:** 3
**Contribution:** 3
**Rating:** 7
**Confidence:** 4

**Summary:**

This paper introduces SSDiff, a model that improves the resolution of satellite imagery by combining high-resolution panchromatic images and low-resolution multispectral images. SSDiff separates image details into spatial and spectral components, and processes them through different network branches, then merges them back together. It utilizes a denoising diffusion model and adopts a designed Alternating Projection Fusion Module (APFM) to complete the spatial-spectral fusion. Additionally, they propose a Frequency Modulation Inter-branch Module (FMIM) to modulate the frequency distribution between branches, as well as a LoRA-like branch-alternating fine-tuning method to further improve performance. The paper provides a solid theoretical foundation, and through experiments on various datasets, SSDiff demonstrates the ability to generate high-resolution multispectral images with minimal spatial and spectral distortion.

**Strengths:**

1. The authors propose an innovative spatial-spectral integrated diffusion model for pansharpening, which is a timely application of diffusion models to this domain.
2. This paper provides a solid theoretical basis for the approach, grounded in concepts from linear algebra and subspace decomposition.
3. The proposed methods in this paper achieve state-of-the-art performance on two datasets. The comprehensive ablation study further adds to the robustness and credibility of the results.
4. This paper is well-organized and the content is clear and easy to read.

**Weaknesses:**

- Efficiency of the method: Evaluating and discussing the efficiency of the DDPM-based method used in SSDiff would help provide a more comprehensive understanding of the practical advantages and limitations of the proposed approach. This information could be valuable for readers to assess the real-world applicability and deployment feasibility of the SSDiff model.
- Experimental Section: The proposed method in this paper has been extensively evaluated for its generalization performance, but the experiments are limited to the Pansharpening task. It would be better if there were experiments on other datasets, but it's not essential.

**Questions:**

1. How does the element-wise multiplication operation help preserve the complementary information from the two domains?
2. Could the author elaborate on any challenges faced when integrating information from PAN and LrMSI into a diffusion model? This clarification would help in understanding your methodology.

**Limitations:**

The authors have provided reasonable Limitations and Broader Impact.

---

> ### Author Rebuttal · Authors · 2024-08-05
>
> >**W1.** Efficiency of the method: Evaluating and discussing the efficiency of the DDPM-based method used in SSDiff would help provide a more comprehensive understanding of the practical advantages and limitations of the proposed approach. This information could be valuable for readers to assess the real-world applicability and deployment feasibility of the SSDiff model.
>
> Thank you for your thorough review and valuable suggestions regarding our manuscript. The diffusion-based method generally has more running time than CNN-based methods due to the multiple timesteps of the diffusion mechanism. The comparison of running time shown in the table below ensures this point. But compared to the Pandiff method, which is also based on a diffusion model, SSDiff still demonstrates a noticeable advantage in terms of efficiency.
>
> |Method|SSDiff|PanDiff|DCFNet|MMNet|LAGConv|
> |:-|:-:|:-:|:-:|:-:|:-:|
> |Runtime (s)|7.417|261.410|0.548|0.348|1.381|
>
> >**W2.** Experimental Section: The proposed method in this paper has been extensively evaluated for its generalization performance, but the experiments are limited to the Pansharpening task. It would be better if there were experiments on other datasets, but it's not essential.
>
> Thank you for your suggestion. Despite conducting many generalization experiments on the pansharpening task, we plan to conduct experiments on datasets from various other tasks in the future to prove the generalization capability of SSDiff across tasks.
>
>
> >**Q1.** How does the element-wise multiplication operation help preserve the complementary information from the two domains?
>
> Thanks for your review. Firstly, after the dot product, the information in $T_{spa}$ and $T_{spe}$ will not be lost. Secondly, in Table 2 of the manuscript, the experiment (i.e., V3, V4, and V5 of the “Effectiveness ..") illustrated the differences when using dot product, concatenation, additional, and multiplication. The result proved that the dot product performed better.
>
> >**Q2.** Could the author elaborate on any challenges faced when integrating information from PAN and LrMSI into a diffusion model? This clarification would help in understanding your methodology.
>
> Thanks for your careful review. We give detailed challenges. The PAN image contains rich high-frequency information and less spectral information, while the LrMSI mainly contains low-frequency information and rich spectral information. The pansharpening aims to fuse these two images to obtain the HrMSI. Besides,  LoRA in conventional DDPM is used to fine-tune with arbitrary domains, whereas our task requires fine-tuning in the spatial and spectral domains. A toy example is shown in Fig.~1. Therefore, it motivates us to propose SSDiff specifically for the pansharpening task. By introducing subspace decomposition into SSDiff, we generalized vector projection to matrix form and designed APFM, which further inspired us to propose the L-BAF.

---

### Official Review · Reviewer_Noom · 2024-07-05

**Soundness:** 3
**Presentation:** 3
**Contribution:** 3
**Rating:** 6
**Confidence:** 4

**Summary:**

For the task of multispectral and panchromatic image fusion (Pansharpening), this paper proposes a spatial-spectral integrated diffusion model (SSDiff). The framework is novel, utilizing spatial and spectral branches to learn spatial details and spectral features separately. It introduces an alternating projection fusion strategy and a frequency modulation inter-branch module to enhance fusion quality. Furthermore, designing a LoRA-like fine-tuning strategy for the proposed dual-branch network to capture component-discriminating features more sufficiently. The study demonstrates promising results across multiple datasets.

**Strengths:**

1. The writing of the article is quite clear.
2. The experiments are conducted quite thoroughly.
3. Subspace decomposition to divide the network into spatial and spectral branches has some degree of innovation. And it has reliable theoretical support
4. The alternating projection fusion strategy and the accompanying LoRA-like fine-tuning method can provide insights for future work.

**Weaknesses:**

1. Figure 4 is not clear. It is hard to see how “ the low-frequency components undergo a gradual modulation characterized by a slow and subtle rate of change in the denoising process. In contrast, the modulation process of the high-frequency components exhibits distinct dynamic variation.”. Suggest enhancing the high-frequency image to make it more prominent.
2. To ensure the fairness of the experiment, it would be reasonable to add a control experiment. Continuing to train the 150k base model for 30k iterations using the fine-tuned model's configuration and then comparing the results to the fine-tuned model. This would provide a more comprehensive evaluation of the effectiveness of the fine-tuning strategy.

**Questions:**

1. What does the first row of data in Table 5 represent? The paper states that "the total training iterations of the WV3 datasets are set to 150k. ... and the total fine-tune training iterations are set to 30k." Is the first row's data the result after training for 150k iterations? If that is the case, please address the issue mentioned concern (see weakness 2).
2. Regarding the limitation mentioned about the efficiency disadvantage of diffusion models, can the authors provide more specific comparative results? The paper noted that diffusion models have an efficiency disadvantage compared to other super-resolution methods. To fully evaluate this limitation, it would be helpful if the authors could include more concrete performance comparisons.
3. Why do low-frequency components undergo progressive modulation while high-frequency components exhibit distinct dynamic variations? I don't quite understand this point.
4. What are the specific differences between this method and other DDPM methods in terms of training and sampling processes, e.g. Meng et al.2023.

**Limitations:**

The authors provide Limitations and Broader Impact, where limitations are influenced by the nature of the task itself.

---

> ### Author Rebuttal · Authors · 2024-08-05
>
> >**W1.** Figure 4 is not clear.
>
> Thank you for your valuable suggestion. We will fix this.
>
>
> >**W2&Q1.** To ensure the fairness of the experiment ..? \& What does the first row of data in Table 5 represent? ..
>
> Thank you for your detailed review. The first row's data is the result after training for 150k iterations. Io ensure the reliability of the L-BAF experiment, the modifications to Table 5 are as follows: the first row represents the evaluation results of the pre-training base model used for fine-tuning, and the second row displays the evaluation results of the experiment conducted with fine-tuning on the pre-training base model (lr=0.0001) for an additional 30k iterations without using the L-BAF method.
>
> |$\mathcal{S}$/$\mathcal{F}$|SAM ($\downarrow$)|ERGAS ($\downarrow$)|Q8 ($\uparrow$)|SCC ($\uparrow$)|
> |:-|:-:|:-:|:-:|:-:|
> |$-$/$-$|2.8646$\pm$0.5241|2.1217$\pm$0.4671|0.9125$\pm$0.0874|0.9863$\pm$0.0040|
> |✘/✘|2.8681$\pm$0.5837|2.1302$\pm$0.5235|**0.9206$\pm$0.0850**|**0.9868$\pm$0.0048**|
> |$\checkmark$/✘|2.8545$\pm$0.5244|2.1138$\pm$0.4658|0.9143$\pm$0.0857|0.9864$\pm$0.0040|
> |✘/$\checkmark$|2.8460$\pm$0.5232|2.1132$\pm$0.4671|0.9152$\pm$0.0849|0.9864$\pm$0.0041|
> |$\checkmark$/$\checkmark$|**2.8429$\pm$0.5284**|**2.1059$\pm$0.4560**|0.9156$\pm$0.0841|0.9867$\pm$0.0038|
>
>
> >**Q2.** Regarding the limitation mentioned about the efficiency disadvantage of diffusion models, can the authors provide more specific comparative results?
>
> Thank you for pointing out this. The diffusion-based method generally has more running time than CNN-based methods due to the multiple timesteps of the diffusion mechanism. The comparison of running time shown in the table below ensures this point. However, $\textbf{for a fair comparison}$ with another diffusion-based method for pansharpening, i.e., PanDiff, our method still gets a significant advantage.
>
> |Method|SSDiff|PanDiff|DCFNet|MMNet|LAGConv|
> |:-|:-:|:-:|:-:|:-:|:-:|
> |Runtime (s)|7.417|261.410|0.548|0.348|1.381|
>
>
> >**Q3.** Why do low-frequency components undergo progressive modulation while high-frequency components exhibit distinct dynamic variations?
>
> Thanks for your review. For a spectral branch, the input LrMSI contains blurred spatial detail and abundant spectral information, which is low-frequency spatial information. Then, we further illustrate the changes in low-frequency and high-frequency information in Fig.~4. During the denoising process, the low-frequency information (e.g., global structure) of images just has some slight changes in the artifacts. However, the high-frequency information (e.g., edges and textures) changes from noisy images to images with edges and textures. Therefore, we designed FMIM to utilize the phenomenon to inject high-frequency information into the spectral branch to enhance changes in frequency information.
>
> >**Q4.** What are the specific differences between this method and other DDPM methods in terms of training and sampling processes, e.g. Meng et al.2023.
>
> Thanks for your review again. Among the existing works based on DDPM, we specifically design a dual-branch structure for the special task, i.e., pansharpening, that essentially fuses two inputs to get one output. Besides, different from the LoRA in conventional DDPM, we propose a novel fine-tuning method (i.e., L-BAF strategy) based on our APFM, which is more suitable for the pansharpening task.

---

### Official Review · Reviewer_kLBw · 2024-07-10

**Soundness:** 3
**Presentation:** 2
**Contribution:** 3
**Rating:** 6
**Confidence:** 4

**Summary:**

This paper propose a novel spatial-spectral integrated diffusion model called SSDiff based on subspace decomposition, and its main contributions contain: 1) Considering existing DDPM-based methods have not yet designed models for the discriminative features required in the pansharpening task, SSDiff divide the network into spatial and branches based on subspace decomposition. Additionally, an alternating projection fusion module (APFM) is constructed for SSDiff to transform the process of fusing HrMSI into the fusion process of spatial and spectral components. Tests on four widely used pansharpening datasets demostrate SSDiff can achieve  state-of-the-art (SOTA) performance. 2) The frequency modulation inter-branch module is used at the junction of spectral and spatial branches to enrich extracted spatial information with more high-frequency information in the denoising process. 3) The proposed L-BAF method is used to fine-tune the network based on the proposed APFM, where the spatial and spectral branches are updated alternately. This design allows alternately fine-tuning the two branches without increasing the parameter count, enabling the learning of more discriminative features.

**Strengths:**

1) Considering existing DDPM-based methods have not yet designed models for the discriminative features required in the pansharpening task,  this paper proposes a novel SSDiff method based on subspace decomposition, which leverages spatial and spectral branches to discriminatively capture global spatial information and spectral features, respectively.

2) And from the perspective of subspace decomposition, an alternating projection fusion module (APFM) is constructed to fuse the captured spatial and spectral components, which is concerned with vector projection.

3) To overcome the problem of uneven distribution of frequency information between two branches in the denoising process, a frequency modulation inter-branch module (FMIM) is designed.

4) Through the proposed LoRA-like branch-wise alternating fine-tuning (L-BAF), SSDiff can further reveal spatial and spectral information not discovered in each branch. And L-BAF is inspired by the idea that subspace decomposition can be generalized into the self-attention mechanism.

5) There are abundant experiments to support the effectiveness of SSDiff from various aspects.

**Weaknesses:**

1) Maybe expression should be pay more attention to, both grammatically and logically. For example, the sentences in line 135 and line 140 are incomplete.

2) This paper claims the fusion result images and error maps of all these methods in Fig. 5, yet it's not the truth. And there lacks concerned descriptions about results on GaoFen-2 full dataset.

3) Paragraph from line 162 to line 167 seems not to be expressed clearly about how eq.(3) and eq.(7) can be generalized into matrix form of the self-attention mechanism as shown in Fig.3.

**Questions:**

1) Why introduce another two features from the features of spatial domain and spectral domain separately?

2) Why choose FFT as the specific way to extract the high-frequency information of the feature map obtained from the spatial branch rather than other methods?

3) The design of FMIM?

**Limitations:**

1) This paper evaluate the effectiveness of the proposed SSDiff over pansharpening task and will extend our method to other multispectral fusion tasks.

2)  The proposed FMIM can adjust the frequency information between the two branches, but it also introduces additional hyperparameters, increasing the difficulty of fine-tuning during training.

3) The time cost for our approach is higher than other DL-based models, primarily due to the limitation imposed by the large number of sampling steps required in the diffusion model.

---

> ### Author Rebuttal · Authors · 2024-08-05
>
> >**Q1.** Why introduce another two features from the features of spatial domain and spectral domain separately?
>
> Thank you for your thorough review and feedback. Firstly, PAN images and LrMSI are obtained from different sensors and contain distinct feature information. PAN images exhibit richer spatial details, while LrMSI possesses more abundant spectral information. Considering the characteristics of the aforementioned pansharpening task, we leverage spatial and spectral branches to discriminatively capture global spatial information and spectral features, respectively. Additionally, we construct an APFM by incorporating the characteristics of the cross-attention mechanism and the task properties. Utilizing the alternate projection method to merge the captured spatial and spectral component features and generate HrMSI.
>
> >**Q2.**  Why choose FFT as the specific way to extract the high-frequency information of the feature map obtained from the spatial branch rather than other methods?
>
> Thank you for your insightful question. The frequency domain representation provided by the FFT aligns well with the intuition that high-frequency information is crucial for capturing spatial details in the pansharpening task. While there may be other methods, such as wavelet-based approaches or high-pass filtering, the FFT-based approach is a common and effective choice due to its efficiency, shift-invariance, and widespread adoption in the field of image processing and fusion.
>
>
> >**Q3.** The design of FMIM?
>
> Thanks for your review again. As shown in Fig.4 in the manuscript. During the denoising process, the low-frequency information (e.g., global structure) of images just has some slight changes in the artifacts. However, the high-frequency information (e.g., edges and textures) changes from noisy images to images with edges and textures. It can be considered that in the process of denoising and reconstructing the HRMSI, the majority of timesteps are mainly spent reconstructing the high-frequency information. Therefore, We designed FMIM to modulate the degree of high-frequency information.  Utilizing a Fourier filter to extract the high-frequency information of the feature map obtained from the spatial branch and inject enhanced high-frequency information into the spectral branch.

---

### Comment · Area_Chair_HWUF · 2024-08-11

Dear Reviewers,
Authors have carefully prepared their rebuttal addressing the concerns you have raised. Please check the rebuttals and join the discussion about the paper.
Thanks,
Your AC

---

> ### Author Response · Authors · 2024-08-14
>
> Dear AC,
> ﻿
> Thank you for your ongoing help and support. We have tried our best to address all concerns of the reviewers. If any new concerns arise, could you please ask the reviewers to inform us?
> ﻿
> Best regards,
> ﻿
> Authors

---

### Author Response · Authors · 2024-08-14

Dear reviewers and AC,
﻿
We want to thank you for your insightful and inspiring feedback. We are very pleased that you consider our work interesting and innovative. Your feedback is incredibly valuable to us, and we are fortunate to have such diligent reviewers contributing to our work. We have addressed each reviewer's comments and questions one by one, and we have also updated some figures and tables.
If we fail to address all the reviewers’ concerns, we sincerely request the reviewers to raise your concerns directly and we will respond as soon as possible.
﻿
Best regards,
Authors

---

### Decision · Program_Chairs · 2024-09-25

**Decision:**

Accept (poster)

**Comment:**

All reviewers found the proposed LoRA -based approach for pansharpening very interesting and novel. Authors presented very good responses to the concerns raised by the reviewers.